# 5-HT3 Antagonist Ondansetron Increases apoE Secretion by Modulating the LXR-ABCA1 Pathway

**DOI:** 10.3390/ijms20061488

**Published:** 2019-03-25

**Authors:** Motoko Shinohara, Mitsuru Shinohara, Jing Zhao, Yuan Fu, Chia-Chen Liu, Takahisa Kanekiyo, Guojun Bu

**Affiliations:** 1Department of Neuroscience, Mayo Clinic, Jacksonville, FL 32224, USA; shinomottyyy@gmail.com (M.S.); Zhao.Jing@mayo.edu (J.Z.); Fu.Yuan@mayo.edu (Y.F.); Liu.chiachen@mayo.edu (C.-C.L.); Kanekiyo.Takahisa@mayo.edu (T.K.); 2Department of Aging Neurobiology, National Center for Geriatrics and Gerontology, Aichi 474-8511, Japan; 3Neuroscience Graduate Program, Mayo Clinic, Jacksonville, FL 32224, USA

**Keywords:** apoE, astrocytes, Alzheimer’s disease, ondansetron, LXR, ABCA1

## Abstract

Apolipoprotein E (apoE) is linked to the risk for Alzheimer’s disease (AD) and thus has been suggested to be an important therapeutic target. In our drug screening effort, we identified Ondansetron (OS), an FDA-approved 5-HT3 antagonist, as an apoE-modulating drug. OS at low micromolar concentrations significantly increased apoE secretion from immortalized astrocytes and primary astrocytes derived from apoE3 and apoE4-targeted replacement mice without generating cellular toxicity. Other 5-HT3 antagonists also had similar effects as OS, though their effects were milder and required higher concentrations. Antagonists for other 5-HT receptors did not increase apoE secretion. OS also increased mRNA and protein levels of the ATB-binding cassette protein A1 (ABCA1), which is involved in lipidation and secretion of apoE. Accordingly, OS increased high molecular weight apoE. Moreover, the liver X receptor (LXR) and ABCA1 antagonists blocked the OS-induced increase of apoE secretion, indicating that the LXR-ABCA1 pathway is involved in the OS-mediated facilitation of apoE secretion from astrocytes. The effects of OS on apoE and ABCA1 were also observed in human astrocytes derived from induced pluripotent stem cells (iPSC) carrying the APOE ε3/ε3 and APOE ε4/ε4 genotypes. Oral administration of OS at clinically-relevant doses affected apoE levels in the liver, though the effects in the brain were not observed. Collectively, though further studies are needed to probe its effects in vivo, OS could be a potential therapeutic drug for AD by modulating poE metabolism through the LXR-ABCA1 pathway.

## 1. Introduction

Alzheimer’s disease (AD) is the most common form of dementia in the elderly population. It is well established that the apolipoprotein E (APOE) gene is the strongest genetic risk factor for AD. Compared to the ε3 allele of APOE (APOE3), the major isoform, the inheritance of one or two ε4 alleles of APOE (APOE4) increases the risk of AD by 3–4-fold and 12–13-fold, respectively [1]. apoE regulates lipid metabolism as a structural component of lipoprotein particles in both the peripheral and the central nervous systems. In the brain, apoE is mainly produced by astrocytes and supplies cholesterol and other lipids to support synaptic formation and function [2]. apoE also plays important roles in neuroinflammation and metabolism of amyloid-β (Aβ). These apoE-related pathways could be interdependently or independently involved in the pathogenesis of AD as well as cognitive function [2,3].

Though it remains unclear how apoE is involved in the pathogenesis of AD, it is of note that the levels of apoE4 are lower than those of apoE3 in the brain of apoE-targeted replacement (TR) mice expressing each of the human apoE isoforms under the control of the endogenous mouse promoter, and in the plasma and cerebrospinal fluid of humans with APOE3 or APOE4 [4,5,6,7]. Moreover, though some controversies exist, recent studies have indicated that liver X receptor (LXR) agonists or retinoid X receptor (RXR) agonists enhance apoE lipidation and secretion through the modulation of the ATB-binding cassette protein A1 (ABCA1), and benefit Aβ metabolism, neuropathology, and cognitive function in animal models of AD [8,9,10,11]. Thus, up-regulation of apoE secretion and lipidation status could be an attractive pharmacological approach for AD therapy. Despite these promises, the potential clinical applications of these compounds are debated due to their intrinsic side effects [12,13].

Thus, to identify novel compounds that can increase apoE levels with a biological safety profile, we performed a drug screen using an FDA-approved drug library, which contains 640 compounds. In the drug screen effort, we identified Ondansetron (OS), an FDA-approved antagonist of the 5-HT3 receptor—a subfamily of receptors for serotonin or 5-hydroxytryptamine (5-HT)—as an apoE-modulating drug. This study also defined the underlying mechanism by which OS regulates apoE secretion using mouse and human astrocyte cultures, and examined the effects of OS on apoE metabolism in vivo using mouse models.

## 2. Results

### 2.1. OS Increases apoE Secretion from Murine Astrocytes

Our initial screening using an FDA-drug library has identified OS as a compound that can significantly increase apoE secretion. To confirm the effects of OS on apoE secretion and test the concentration dependency, we treated immortalized astrocytes from apoE3-TR mice with different doses of OS for 24 h and observed that 1 μM or higher doses of OS increased apoE levels in a concentration-dependent manner (Figure 1A). Importantly, these doses of OS did not affect cell viability, measured by the 3-(4,5-dimethylthiazol-2-yl)-2,5-diphenyl tetrazolium bromide (MTT) assay (Figure 1B). We also treated primary astrocytes from apoE3-TR mice with OS to confirm that its effects on apoE secretion did not depend on immortalized cell lines. Indeed, OS increased apoE secretion (Figure 1C) without affecting cell viability (Figure 1D) in these primary astrocytes. Of note, the effects of OS in these primary astrocytes were observed at one order of magnitude lower than that needed for immortalized cells, suggesting more potent effects to increase apoE secretion in vivo. OS also increased apoE secretion from immortalized astrocytes from apoE4-TR mice, indicating that OS effects did not depend on apoE isoform (Figure 1E).

Then, we checked whether other 5-HT3 antagonists or compounds targeting any other 5-HT receptors had similar effects on apoE secretion. In these experiments, we used OS as a positive control (Figure 2A). Although one 5-HT3 antagonist, Granisetron, did not affect apoE levels, two other 5-HT3 antagonists, Dolasetron and Tropisetron, increased apoE levels at 10 μM (Figure 2E–G). However, compounds targeting other 5-HT receptors, Methysergid (5-HT1/2 antagonist), Ketanserin (5-HT2 antagonist), Clozapine (5-HT2 antagonist), GR-113,808 (5-HT4 antagonist), SB-699,551 (5-HT5 antagonist), SB-258,585 (5-HT6 antagonist), and SB-269,970 (5-HT7 antagonist) did not increase apoE secretion (Figure 2B–D,H–K). Additionally, serotonin itself did not affect apoE secretion (Figure 2L). These results suggest that the effects of OS are specific to 5-HT3 receptors.

### 2.2. OS Increases apoE Secretion through LXR-ABCA1 Pathway

Next, we studied the mechanism underlying how OS increases apoE secretion. As the effects of several compounds on promoting apoE secretion are mediated by increasing ABCA1 expression [14], we assessed the ABCA1 levels after treatment with OS. Interestingly, OS increased the ABCA1 protein levels in apoE3 immortalized astrocytes (Figure 3A). On the other hand, OS did not affect protein levels of ABCG1, low-density lipoprotein receptor (LDLR), and the LDLR-related protein 1 (LRP1) (Figure 3B–D). Although OS did not apparently affect the apoE mRNA levels, ABCA1 mRNA levels were also significantly increased by OS treatment (Figure 3E,F), suggesting an analogous effect to LXR or RXR agonists [8,14]. We also confirmed that T0901317, an LXR agonist, had similar effects on levels of apoE, LDLR, LRP1, ABCG1, and ABCA1 to OS (Shinohara et al., unpublished work, 2019). Moreover, OS increased the secretion of higher molecular size, potentially lipidated apoE, consistent with an involvement of the ABCA1 pathway (Figure 3G) [14].

Importantly, geranylgeranyl pyrophosphate (GGPP), an LXR antagonist [15,16], blocked the effects of OS on apoE secretion and ABCA1 expression (Figure 4A,B). Consistently, cyclosporin A, an ABCA1 antagonist [17,18], blocked the effects of OS on apoE secretion (Figure 4C). Taken together, these results indicate that OS increases apoE secretion through the LXR-ABCA1 pathway.

### 2.3. OS Increases apoE Secretion from Human Astrocytes

To confirm the effects of OS on apoE secretion in human astrocytes, we introduced human astrocytes derived from iPSC. Indeed, OS increased apoE secretion in ε3/ε3 human astrocytes and ε4/ε4 human astrocytes in a concentration-dependent manner (Figure 5A,D). Of note, the effects of OS on apoE secretion in these human astrocytes were observed at 0.1 μM, similar to mouse primary astrocytes. OS treatment also increased ABCA1 levels, but not ABCG1 levels, in both ε3/ε3 human astrocytes (Figure 5B,C) and ε4/ε4 human astrocytes (Figure 5E,F), consistent with the results of mouse astrocyte experiments. These results indicate that OS can increase apoE secretion from human astrocytes irrespective of apoE isoforms.

### 2.4. Effects of OS on apoE Levels In Vivo

To address the in vivo relevance, we assessed the effects of OS in mice. We treated apoE3-TR mice daily with OS intraperitoneal (i.p.) at 1 mg/kg/day, which was consistent with a previous study treating neurological symptoms [19,20,21], or gradually increased the doses (3 mg/kg/day and 10 mg/kg/day) for seven days. With these doses, OS did not increase aspartate aminotransferase (AST) levels, a marker of liver toxicity, while a higher dose (100 mg/kg/day) was highly toxic and lethal (Shinohara et al., unpublished work, 2019). In this experimental scheme, OS did not affect apoE levels in the brains even at the higher doses (Figure 6A), while apoE levels were increased in the liver of mice treated with 1 mg/kg/day and 3 mg/kg/day of OS (Figure 6B).

## 3. Discussion

Although up-regulation of apoE secretion via the ABCA1 pathway is an attractive target for AD and other age-related neurological disorders, the clinical application of modulators of this pathway has not yet been established, partly because of their side effects [12,13]. Therefore, drugs with safer profiles and long-term tolerance are needed for the treatment and prevention of AD. Thus, we conducted drug screening using an FDA-approved drug library and identified OS as a modulator of apoE secretion with a safety profile in cultured astrocytes. Importantly, our results showed that the effects of OS are more evident in murine primary astrocytes and human astrocytes where OS could increase apoE secretion in the nanomolar range. Moreover, we found that OS could also increase high molecular size apoE particles and ABCA1 levels, and these effects were blocked by LXR or ABCA1 antagonists. Clinically relevant doses of OS also increased apoE levels in the liver, though OS effects in the brain were not observed.

OS is a 5-HT3 receptor antagonist, used for the treatment of nausea and vomiting associated with chemotherapy or pregnancy [22]. As we found that other 5-HT3 receptor antagonists also had similar (but weaker) effects on apoE secretion, this effect could be mediated by antagonism of the 5-HT3 receptor. However, it is not yet clear why OS showed stronger effects compared to the other three 5-HT3 receptor antagonists with similar or higher binding affinity to the 5-HT3 receptor [23]. The chemical structure of OS is different from the other three 5-HT3 receptor antagonists; OS is categorized as a carbazole derivative, while the other three antagonists are categorized as indazoles or indoles. Such distinct chemical structures result in different metabolism by cytochrome P450 families, and potentially different effects [24]. Indeed, independently of the antagonistic properties of OS at the 5-HT3 receptor, OS can attenuate the inositol 1,4,5-trisphosphate signaling pathway [25], which could be important for the peroxisome proliferator-activated receptor α-mediated inhibition of the LXR-ABCA1 pathway [26]. Thus, the inhibition of such specific pathways by OS might be involved in the increase in apoE secretion via the LXR-ABCA1 pathway. It warrants further investigation regarding how OS strongly affects the LXR-ABCA1 pathway among the 5-HT3 receptor antagonists.

Regarding the lipidation status, while we observed that OS increased apoE particles in the large molecular size fractions (Figure 3G), corresponding to the fractions where most cholesterol existed and its levels were increased by OS (Figure 3G: inset), we did not determine how much lipidated and non-lipidated apoE was altered by OS. Though it is natural to think OS increases lipidated apoE rather than non-lipidated apoE via the ABCA1 pathway [9,27], further studies would be necessary to confirm such a hypothesis.

The main clinical effects of OS in vivo are carried out via the peripheral vagus nerve [28], thus there has not been a careful examination of whether and how much of OS can penetrate the blood-brain barrier (so-called BBB permeability) and affect cells in the brain. Simpson et al. reported that after a single dose, OS levels in the CSF (2.6–15.4 ng/mL) reached 15% of that in the plasma (39.5–147 ng/mL), indicating a potentially low BBB permeability of OS [29]. According to several studies, OS could be effective in the treatment of some neurological symptoms associated with psychiatric problems or dyskinesia [30,31,32]; therefore, it may be that such low levels of OS in the brain could exert at least partial beneficial effects. However, the failure of OS to increase apoE levels in the brain indicates that pharmacological modification to increase BBB permeability would be necessary to achieve target effects on ABCA1 and apoE in the brain.

On the other hand, OS increased apoE levels in the liver, indicating that OS can affect peripheral ABCA1-apoE metabolism. This result is consistent with a previous study showing that plasma lipoprotein levels were up-regulated with OS treatment [33]. Although a previous clinical study failed to see improvement of cognitive deficits in AD patients with OS [34], it is of note that recent studies have shown that peripheral apoE metabolism could play an important role in the pathogenesis of AD; individuals with low levels of plasma apoE were more likely to develop AD in the future [6,7]. Thus, peripheral apoE metabolism could also be an important target for the treatment of AD. In this regard, OS might be effective for preventing AD through the modulation of peripheral apoE metabolism.

In summary, we found that OS increased apoE secretion from cultured murine and human astrocytes through the LXR-ABCA1 pathway. Although we failed to observe the effect of OS in the brain, we did observe significant effects of OS on peripheral apoE metabolism. Thus, further studies are warranted to interrogate potential effects of OS in AD and AD-related pathways.

## 4. Materials and Methods

### 4.1. Cell Lines and Reagents

Immortalized apoE3 astrocytes were a kind gift from Dr David M. Holtzman (Washington University in St. Louis, MO, USA) [14]. The screened compounds were from a commercially available FDA-approved drug library (Enzo Life Sciences, Farmingdale, NY, USA) containing 630 compounds. OS, GGPP, T0901317, cyclosporin A, and puromycin were purchased from Sigma-Aldrich (St. Louis, MO, USA). GR-113808, SB-699551, SB-258585, and SB-269970 were purchased from R&D (Minneapolis, MN, USA). All compounds were dissolved into 10 mM stock solution in dimethyl sulfoxide (DMSO). The mouse monoclonal anti-ABCA1 (ab18180) was purchased from Abcam (Cambridge, MA, USA) and the mouse monoclonal anti-actin antibody was purchased from Sigma (A5316).

### 4.2. Immortalized Astrocyte Cultures

Immortalized apoE3 or apoE4 astrocytes were cultured as previously described [14]. In brief, cells were cultured in astrocyte medium (DMEM/F12 (Thermo Fisher, Waltham, MA, USA), 20% fetal bovine serum, 2 mM l-glutamine, glutamine, 1× non-essential amino acids, 1 mM sodium pyruvate and 1% penicillin/streptomycin) at 37 °C in humidified air containing 5% CO_2_. For primary screening and dose-dependent experiments, immortalized astrocytes were seeded and cultured overnight in 96-well plates at 10,000 cells/well density. For Western blotting and RT-qPCR analyses, cells were seeded the day before treatments in 6-well plates (400,000 cells/well) and then treated with selected compounds or vehicle (DMSO) in serum-free astrocyte medium.

### 4.3. Primary Astrocyte Cultures

Primary astrocytes were prepared from new-born (P1–P2) human apoE3-TR mice as described previously [14]. In brief, the brain was removed from the skull and the meninges were discarded. Subsequently, the cortices were minced and incubated with 0.05% trypsin in a water bath at 37 °C for 15 min. Enzyme-digested dissociated cells were triturated with astrocyte growth medium and centrifuged at 300× *g* for five minutes. The pellet was resuspended, passed through a 70 μm nylon mesh, washed, and centrifuged at 300× *g* for five minutes. The cells were plated on poly-d-lysine–coated 75 cm^2^-flask in astrocyte growth media, which were changed every three days. Cells were grown until confluence and then re-plated for the experiments at a density of 2 × 10^5^ cells/well on 12-well plates coated with Poly-d-Lysine.

### 4.4. Human Astrocyte Cultures

Human astrocytes were prepared from induced pluripotent stem cells (iPSCs) derived from cognitively normal individuals with different APOE genotypes (ε3/ε3 or ε4/ε4) as described previously [35]. Briefly, iPSCs were digested with Dispase (Stemcell Technologies, Vancouver, BC, Canada) for 5 min, then iPSC clumps were washed down and cultured in neural induction medium in suspension for 5–7 days for neural differentiation. Next, neurospheres were seeded onto Matrigel-coated dishes to induce neural rosette formations. Neural rosettes were manually picked, digested into single cells with Accutase (Stemcell Technologies), and re-plated onto Matrigel-coated dishes in a neural induction medium for another 10–14 days. Neural progenitor cells (NPCs) were amplified in neural progenitor cell medium (Stemcell Technologies). Astrocyte differentiation was started by changing the medium into astrocyte differentiation medium, composed of astrocyte medium (ScienCell, Carlsbad, CA, USA) with CNTF (10 ng/mL), BMP4 (10 ng/mL), and Heregulin (10 ng/mL) (all from Stemcell Technologies). Cells were passaged during the differentiation process when they reached 80% confluence. Astrocytes at day 40 were used for further drug treatment experiments.

### 4.5. Biochemical Analyses

For Western blotting, samples were lysed in a radio-immunoprecipitation assay (RIPA, Millipore, Billerica, MA, USA) buffer containing protease inhibitor cocktail (Roche, Florence, SC, USA), and analyzed by SDS-PAGE as previously described [14]. Levels of apoE and LRP1 were measured by ELISA as previously described [14,36]. LDLR ELISA was performed by using the Irene capture antibody [37] and the biotin-conjugated goat anti-mouse LDLR detector antibody (R&D). The recombinant mouse LDLR protein (R&D) was used as a standard. For mRNA analysis, total RNA was isolated using the RNeasy Mini Kit (QIAGEN, Germantown, MD, USA) and subjected to DNase I digestion, reverse transcription (SuperScript II RNase H-reverse transcriptase: Life Technologies Corporation, Grand Island, NY, USA), and quantitative real-time PCR to measure levels of ABCA1, apoE, and β-actin. The CT method was used to calculate the relative amount of target genes in each group, as previously described [14]. Size-exclusion chromatography of secreted apoE was performed as previously described [14]. In brief, conditioned medium was concentrated 60-fold with a 10-kDa cut-off filter (Millipore Corporation, Bedford, MA, USA), and passed through tandem Superose-6, 10/300 GL columns (GE Healthcare, Chicago, IL, USA) in PBS containing 50 mM sodium phosphate, PH 7.4, with 150 mM NaCl, 1 mM EDTA, and 0.02% sodium azide at a flow rate of 0.4 mL/min by fast performance liquid chromatography (FPLC). Fractions were collected and analyzed for apoE and cholesterol levels by apoE ELISA and Amplex Red Cholesterol Assay kit (Thermo Fisher), respectively.

### 4.6. In Vivo Effects of Tested Compounds

Male apoE3-TR mice (7–8 months old) were treated (i.p., daily) with OS or vehicle as a control for one week. OS was solubilized in PBS plus hydrochloride (pH > 5.0). Four to six hours after the last administration, mice were transcardially perfused with PBS, and then the brain and liver tissue were dissected and kept frozen at −80 °C until analysis. All animal procedures were approved (A70512) by the Institutional Animal Care and Use Committee (IACUC) at the Mayo Clinic and in accordance with the regulations of the American Association for the Accreditation of Laboratory Animal Care.

### 4.7. Cytotoxicity Assays

In vitro cytotoxic effects of compounds were evaluated with MTT assay kits (Roche) following the manufacturer’s instructions, using puromycin treatment (10 μg/mL) as a positive control. The in vivo cytotoxic effects were evaluated by the AST activity assay kit (Sigma) according to the manufacturer’s instructions.

### 4.8. Statistical Analyses

All quantified data were analyzed by an analysis of variance (ANOVA) with post-hoc Tukey-Kramer test by using JMP (version 10.0, SAS Institute Inc., Cary, NC, USA). Error bars represent standard deviation and *p* < 0.05 was considered significant.

## Figures and Tables

**Figure 1 ijms-20-01488-f001:**
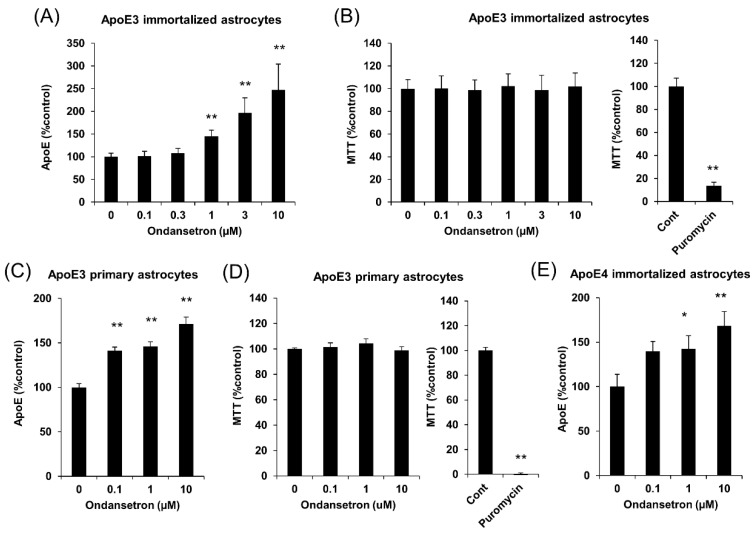
OS increases apoE secretion in immortalized astrocytes or primary astrocytes derived from apoE3-TR mice and apoE4-TR mice. (**A**,**B**) Immortalized astrocytes derived from apoE3-TR mice were cultured and treated with the indicated concentrations of OS. After treatment with OS for 24 h, apoE levels in culture media were determined by ELISA (**A**), and cell viability was assessed by MTT assay (**B**, left). Puromycin treatment was used as a positive control for MTT assay (**B**, right). (**C**,**D**) Primary astrocytes derived from apoE3-TR mice were cultured and treated with the indicated concentrations of OS. After treatment with OS for 24 h, apoE levels in culture media were determined by ELISA (**C**), and cell viability was assessed by MTT assay (**D**, left). Puromycin treatment was used as a positive control for MTT assay (**D**, right). apoE levels in culture media of immortalized astrocytes derived from apoE4-TR mice (**E**) were determined by ELISA after treatment with indicated concentrations of OS for 24 h. Data represent mean ± SD ((**A**,**B**): *n* = 6, biological replicate group, (**C**,**D**): *n* = 3, biological replicate group). * *p* < 0.05, ** *p* < 0.01; Tukey-Kramer test.

**Figure 2 ijms-20-01488-f002:**
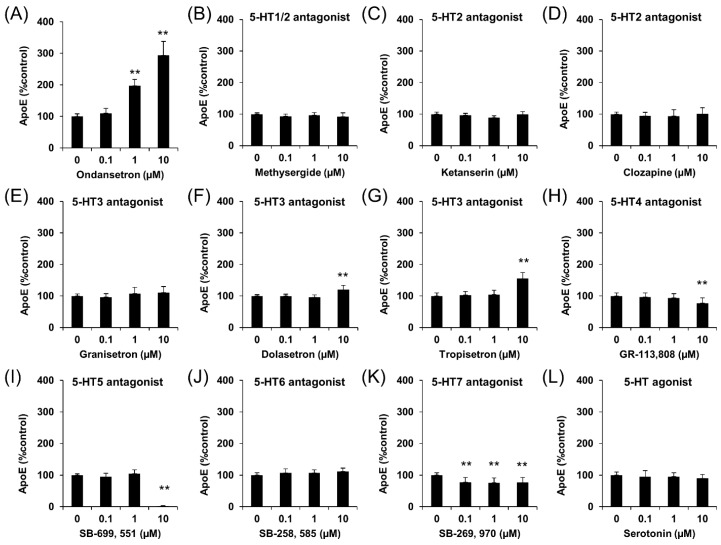
Effects of different 5-HT receptor antagonists on apoE secretion in immortalized astrocytes. apoE levels in culture media of immortalized astrocytes derived from apoE3-TR mice were determined by ELISA after treatment for 24 h with the indicated concentrations of (**A**) OS, (**B**) Methysergide, (**C**) Ketanserin, (**D**) Clozapine, (**E**) Granisetron, (**F**) Dolasetron, (**G**) Tropisetron, (**H**) GR-113,808, (**I**) SB-699,551, (**J**) SB-258,585, (**K**) SB-269,970, and (**L**) Serotonin. Data represent mean ± SD (*n* = 4, biological replicate group). ** *p* < 0.01; Tukey-Kramer test.

**Figure 3 ijms-20-01488-f003:**
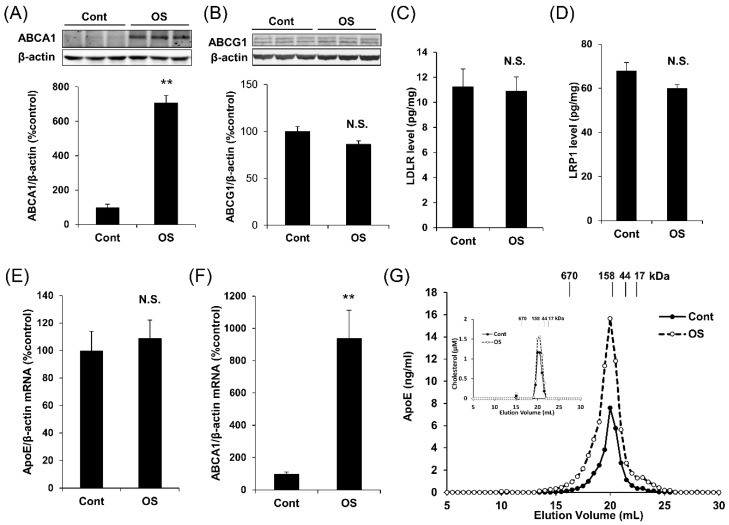
OS increases ABCA1 expression and high molecular size apoE secretion. Protein levels of ABCA1 (**A**) and ABCG1 (**B**) in immortalized astrocytes derived from apoE3-TR mice were analyzed by Western blotting after treatment with OS (1 μM) for 24 h. The graph represents the quantification of ABCA1 and ABCG1 levels. (**C**–**E**) After treatment with OS (1 μM) for 24 h, protein levels of LDLR (**C**) and LRP1 (**D**) were determined by ELISA, and mRNA levels of apoE (**E**) and ABCA1 (**F**) were measured by qRT-PCR and normalized by the mRNA levels of β-actin in immortalized astrocytes derived from apoE3-TR mice. (**G**) The culture media were concentrated and then fractioned by size-exclusion chromatography on fast protein liquid chromatography (FPLC) using tandem Superose-6 columns. The concentration of apoE and cholesterol (inset) in each fraction was analyzed by ELISA and enzymatic assay, respectively. (**A**–**F**) Data represent mean ± SE (**A**–**F**: *n* = 3, biological replicate group). ** *p* < 0.01, NS; non-significant; Student *t*-test.

**Figure 4 ijms-20-01488-f004:**
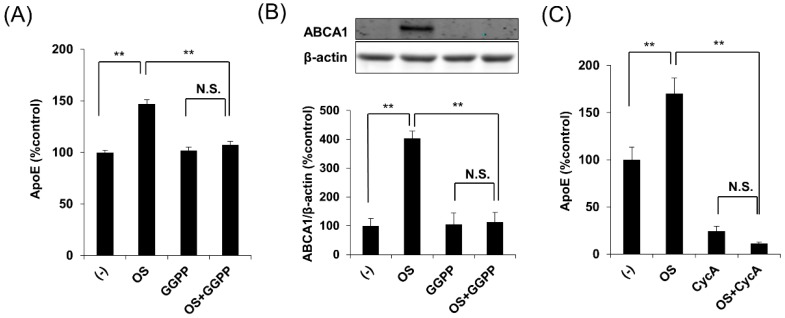
LXR-ABCA1 antagonists block the OS-induced increase in apoE secretion. (**A**) apoE levels in culture media and (**B**) cellular ABCA1 levels in immortalized astrocytes derived from apoE3-TR mice were determined by ELISA (**A**) or Western blotting (**B**) after treatment for 24 h with OS (1 μM) in the absence or presence of an LXR antagonist, GGPP (10 μM). (**C**) apoE levels in culture media of immortalized astrocytes derived from apoE3-TR mice were determined by ELISA after treatment for 24 h with OS (1 μM) in the absence or presence of an ABCA1 antagonist, cyclosporin A (CycA, 10 μM). Data represent mean ± SE (**A**,**C**: *n* = 4, biological replicate group, **B**: *n* = 3, biological replicate group). ** *p* < 0.01, NS; non-significant; Tukey-Kramer test.

**Figure 5 ijms-20-01488-f005:**
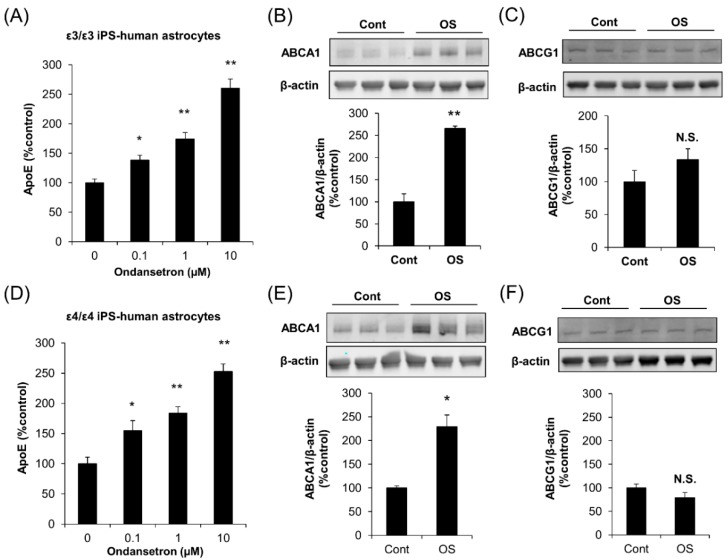
OS increases apoE secretion in astrocytes derived from human-induced pluripotent stem cell. Human iPSC-astrocytes with APOE ε3/ε3 (**A**–**C**) and APOE ε4/ε4 (**D**–**F**) genotypes were cultured and treated with the indicated concentrations of OS for 24 h, and apoE levels in culture media were determined by ELISA (**A**,**D**), and cellular ABCA1 (**B**,**E**) and ABCG1 (**C**,**F**) levels were analyzed by Western blotting using β-actin as a loading control. Data represent mean ± SD (*n* = 3, biological replicate group). * *p* < 0.05, ** *p* < 0.01, NS; non-significant; Tukey-Kramer test.

**Figure 6 ijms-20-01488-f006:**
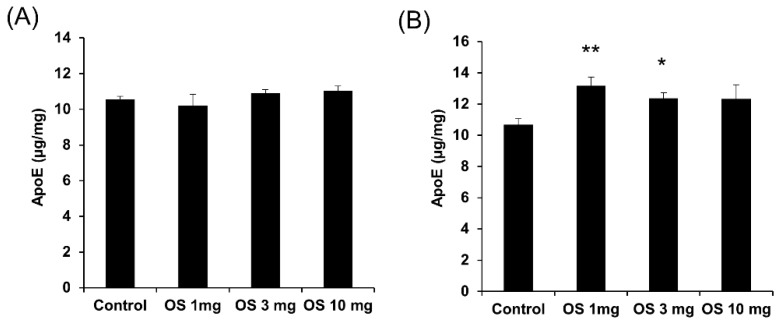
Effects of OS in vivo. apoE levels in the brain (**A**) and the liver (**B)** of apoE3-TR mice (male, four months old) were determined by ELISA after one week of daily intraperitoneal (i.p.) administration of OS (1 mg/kg/day, 3 mg/kg/day, or 10 mg/kg/day). Data represent mean ± SE (*n* = 4, biological replicate group). * *p* < 0.05, ** *p* < 0.01; Tukey-Kramer test.

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
