# Peer review of "5-HT3 Antagonist Ondansetron Increases apoE Secretion by Modulating the LXR-ABCA1 Pathway"

_ijms, 2019, doi:10.3390/ijms20061488_

Round 1
Reviewer 1 Report
In their manuscript entitled “5-HT3 antagonist Ondansetron increases apoE3 secretion by modulating the LXR-ABCA1 pathway”, Shinohara et al report the effect of an antagonist of 5-HT3 receptor (Ondansetron, OS) on apoE secretion from immortalized astrocytes and primary astrocytes. OS also increased mRNA and protein levels of ABCA1. The effects of OS on apoE and ABCA1 were also observed in human astrocytes derived from induced pluripotent stem cells (iPSC) carrying APOE ε3/ε3 and APOE ε4/ε4 genotypes.
The authors attempt to define the underlying mechanism by which OS regulates apoE secretion. They direct their attention to earlier findings that LXR/RXR agonists enhance ABCA1-mediated apoE secretion/lipidation to see if OS treatment would alter ABCA1-mediated apoE secretion and lipidation.
While the concept is interesting to the audience, there are several major concerns:
The authors show that OS increased apoE secretion in apoE3-immortalized astrocytes and primary astrocytes from apoE3-TR mice, and that OS increased ABCA1protein expression levels and mRNA levels; however, it is intriguing that there were no changes in apoE mRNA levels. What about intracellular levels of apoE protein?
There is no direct evidence of lipidation of apoE3 to support their conclusion that OS treatment results in increased lipidated apoE3 via the ABCA1 pathway. From Figure 3G, which shows size exclusion chromatography of concentrated conditioned medium from apoE3 immortalized astrocytes, it appears that the protein appearing around 20 ml (~150 kDa) likely corresponds to tetrameric lipid-free apoE3 (apoE is ~ 34 kDa). It is well known that lipid-free apoE3 self-associates and exists as a mixture of dimers, tetramers and higher state oligomers.
To demonstrate presence of lipidated apoE3, it is critical to show the presence of lipids and/or the lipid composition. In addition, show the size of the lipidated species by other techniques such as non-denaturing PAGE (or perhaps MALS, analytical centrifugation).
It is most likely that secreted apoE3 remains bound to cell surface heparan sulfate proteoglycans, especially if it is lipid-free or poorly lipidated. The authors should determine if secreted apoE3 is lipid-free, poorly/partially lipidated or HSPG-bound.
Although they show that OS also increased secretion of apoE4, the isoform associated with Alzheimer’s disease, a bulk of their experiments were focused apoE3. It is important to show that OS increases secretion of lipidated apoE4.
The in vivo relevance of OS response is not clear, especially due to lack of response in apoE levels in brain of apoE3-TR mice treated with OS.
Minor comments:
There are several grammatical errors in the text.
Citation of Figure 2B-D should be Figure 2F-G in text
The text is written in a sloppy manner with notes such as “This section may be divided by subheadings. It should provide a concise and precise description of the experimental results, their interpretation as well as the experimental conclusions that can be drawn” and “Authors should discuss the results and how they can be interpreted in perspective of previous studies and of the working hypotheses. The findings and their implications should be discussed in the broadest context possible. Future research directions may also be highlighted” still retained in the final submitted version of the manuscript.
Author Response
Comments and Suggestions for Authors
In their manuscript entitled “5-HT3 antagonist Ondansetron increases apoE3 secretion by modulating the LXR-ABCA1 pathway”, Shinohara et al report the effect of an antagonist of 5-HT3 receptor (Ondansetron, OS) on apoE secretion from immortalized astrocytes and primary astrocytes. OS also increased mRNA and protein levels of ABCA1. The effects of OS on apoE and ABCA1 were also observed in human astrocytes derived from induced pluripotent stem cells (iPSC) carrying APOE ε3/ε3 and APOE ε4/ε4 genotypes.
The authors attempt to define the underlying mechanism by which OS regulates apoE secretion. They direct their attention to earlier findings that LXR/RXR agonists enhance ABCA1-mediated apoE secretion/lipidation to see if OS treatment would alter ABCA1-mediated apoE secretion and lipidation.
While the concept is interesting to the audience, there are several major concerns:
The authors show that OS increased apoE secretion in apoE3-immortalized astrocytes and primary astrocytes from apoE3-TR mice, and that OS increased ABCA1protein expression levels and mRNA levels; however, it is intriguing that there were no changes in apoE mRNA levels. What about intracellular levels of apoE protein?
--- We appreciate the reviewer’s comments. As the reviewer indicated, we observed no changes in apoE mRNA levels despite increase in apoE secretion and ABCA1 protein expression. As our primary outcome was to see apoE secretion and ABCA1 protein expression, we did not have data regarding intracellular levels of apoE protein. We are planning to perform to measure intracellular levels of apoE protein in the future.
There is no direct evidence of lipidation of apoE3 to support their conclusion that OS treatment results in increased lipidated apoE3 via the ABCA1 pathway. From Figure 3G, which shows size exclusion chromatography of concentrated conditioned medium from apoE3 immortalized astrocytes, it appears that the protein appearing around 20 ml (~150 kDa) likely corresponds to tetrameric lipid-free apoE3 (apoE is ~ 34 kDa). It is well known that lipid-free apoE3 self-associates and exists as a mixture of dimers, tetramers and higher state oligomers.
To demonstrate presence of lipidated apoE3, it is critical to show the presence of lipids and/or the lipid composition. In addition, show the size of the lipidated species by other techniques such as non-denaturing PAGE (or perhaps MALS, analytical centrifugation).
--- We appreciate the reviewer’s comments. With respect to the reviewer, however, we did not conclude that OS treatment results increased lipidated apoE via the ABCA1 pathway, but we concluded that OS treatment results increased apoE secretion via the ABCA1 pathway. Nonetheless, we indeed measured cholesterol levels in the fractions obtained by size-exclusion chromatography and observed that fractions abundant in cholesterols corresponds with fractions abundant in apoE, and OS also increased cholesterols in such fractions (see below data). This data at least suggests the presence of cholesterols in the fraction where OS increased apoE levels. However, as this observation still would be not enough to conclude that OS increased lipidated apoE (as the reviewer might agree), we would like to conclude like that. We have added discussions in the revised manuscript to clarify our observation and conclusions regarding this point (line 194-198).
It is most likely that secreted apoE3 remains bound to cell surface heparan sulfate proteoglycans, especially if it is lipid-free or poorly lipidated. The authors should determine if secreted apoE3 is lipid-free, poorly/partially lipidated or HSPG-bound.
--- We appreciate the reviewer’s comments. With respect to the reviewer, however, our conclusion in this study is that OS treatment results increased apoE secretion via the ABCA1 pathway, irrespective of lipidation status. We are planning to perform such studies in the future.
Although they show that OS also increased secretion of apoE4, the isoform associated with Alzheimer’s disease, a bulk of their experiments were focused apoE3. It is important to show that OS increases secretion of lipidated apoE4.
--- We appreciate the reviewer’s comments. With respect to the reviewer, however, our conclusion in this study is that OS treatment results increased apoE secretion via the ABCA1 pathway, irrespective of lipidation. We are planning to perform such studies in the future.
The in vivo relevance of OS response is not clear, especially due to lack of response in apoE levels in brain of apoE3-TR mice treated with OS.
--- We appreciate the reviewer’s comments. We have discussed limitation and potential importance of our in vivo findings in the manuscript (line 199-220).
Minor comments:
There are several grammatical errors in the text.
--- We appreciate the reviewer’s comments. We have checked entire manuscript and changed grammatical errors in the revised manuscript.
Citation of Figure 2B-D should be Figure 2F-G in text
--- We appreciate the reviewer’s comments. We have changed citation of these figures, accordingly.
The text is written in a sloppy manner with notes such as “This section may be divided by subheadings. It should provide a concise and precise description of the experimental results, their interpretation as well as the experimental conclusions that can be drawn” and “Authors should discuss the results and how they can be interpreted in perspective of previous studies and of the working hypotheses. The findings and their implications should be discussed in the broadest context possible. Future research directions may also be highlighted” still retained in the final submitted version of the manuscript.
--- We appreciate the reviewer’s comments. We have removed these notes, accordingly.
Reviewer 2 Report
This is a well written manuscript describing interesting results on the effect of OS on ApoE produced by immortalized astrocytes and alive animals. As the authors recognized, it seems there is not significant effect of OS administration on the brain, posing so doubts on a possible applicability in clinical real-life setting. However, the paper is acceptable for publications to me since it offers possible important points of future investigations in AD treatment based on the possible interaction of other types of drugs on ApoE secretion (assuming that the secretion of different levels of ApoE is indeed part of the AD dementia pathogenesis).
1) Could the authors propose a more detailed explanation/hypothesis why OS seems to work better than other 5-HT3 antagonists compunds?
2) Lines 162-164: it seems that the statement "This section may be divided....." does not belong to the rest of the text.
Author Response
Comments and Suggestions for Authors
This is a well written manuscript describing interesting results on the effect of OS on ApoE produced by immortalized astrocytes and alive animals. As the authors recognized, it seems there is not significant effect of OS administration on the brain, posing so doubts on a possible applicability in clinical real-life setting. However, the paper is acceptable for publications to me since it offers possible important points of future investigations in AD treatment based on the possible interaction of other types of drugs on ApoE secretion (assuming that the secretion of different levels of ApoE is indeed part of the AD dementia pathogenesis).
--- We appreciate the reviewer’s positive comments on our manuscript.
1) Could the authors propose a more detailed explanation/hypothesis why OS seems to work better than other 5-HT3 antagonists compunds?
--- We appreciate the reviewer’s comments. We have added more explanation why OS seems to work better than other 5-HT3 antagonist compounds in the discussion of the revised manuscript (line 185-193).
2) Lines 162-164: it seems that the statement "This section may be divided....." does not belong to the rest of the text.
--- We appreciate the reviewer’s comments. We have removed these notes, accordingly.
Round 2
Reviewer 1 Report
The authors determined cholesterol levels in the size exclusion chromatography fractions and showed a small (significant?) increase in cholesterol levels compared to control. If this result can be replicated it is worth showing the data instead of indicating 'data not shown'.